# Plum-Derived Exosome-like Nanovesicles Induce Differentiation of Osteoblasts and Reduction of Osteoclast Activation

**DOI:** 10.3390/nu15092107

**Published:** 2023-04-27

**Authors:** Yu-Seong Park, Hyun-Woo Kim, Jin-Hyeon Hwang, Jung-Young Eom, Dong-Ha Kim, Jinho Park, Hyun-Jin Tae, Seunghoon Lee, Jae-Gyu Yoo, Jee-In Kim, Jae-Hwan Lim, In-Sook Kwun, Moon-Chang Baek, Young-Eun Cho, Do-Kyun Kim

**Affiliations:** 1Department of Molecular Medicine, Cell and Matrix Research Institute (CMRI), School of Medicine, Kyungpook National University, Daegu 41944, Republic of Korea; 2Korea Zoonosis Research Institute, Jeonbuk National University, Iksan 54531, Republic of Korea; 3College of Veterinary Medicine, Jeonbuk National University, Iksan 54596, Republic of Korea; 4Department of Veterinary Anatomy, College of Veterinary Medicine and Institute of Animal Transplantation, Jeonbuk National University, Iksan 54596, Republic of Korea; 5Department of Animal Biotechnology Division, National Institute of Animal Science, Rural Development Administration, Wanju 55365, Republic of Korea; 6Department of Biochemistry and Cell Biology, School of Medicine, Kyungpook National University, Daegu 41944, Republic of Korea; 7Department of Biological Science, Andong National University, Andong 36729, Republic of Korea; 8Department of Food and Nutrition, Andong National University, Andong 36729, Republic of Korea; iskwun@anu.ac.kr

**Keywords:** plum-derived exosome-like nanovesicles (PENVs), exosome-like nanovesicles, osteoblasts, osteoclasts, bone remodeling, BMP-2 signaling, Runx2

## Abstract

Osteoblasts and osteoclasts play crucial roles in bone formation and bone resorption. We found that plum-derived exosome-like nanovesicles (PENVs) suppressed osteoclast activation and modulated osteoblast differentiation. PENVs increased the proliferation, differentiation, and mineralization of osteoblastic MC3T3-E1 cells and osteoblasts from mouse bone marrow cultures. Notably, PENVs elevated the expression of osteoblastic transcription factors and osteoblast differentiation marker proteins in MC3T3-E1 cells. Higher levels of phosphorylated BMP-2, p38, JNK, and smad1 proteins were detected in PENV-treated MC3T3-E1 cells. Additionally, the number of TRAP-positive cells was significantly decreased in PENV-treated osteoclasts isolated from osteoblasts from mouse bone marrow cultures. Importantly, osteoclastogenesis of marker proteins such as PPAR-gamma, NFATc1, and c-Fos were suppressed by treatment with PENVs (50 μg/mL). Taken together, these results demonstrate that PENVs can be used as therapeutic targets for treating bone-related diseases by improving osteoblast differentiation and inhibiting osteoclast activation for the first time.

## 1. Introduction

Bone remodeling is regulated by achieving a balance between healthy new bone formation by osteoblasts and disassembly of old bone by osteoclasts [1,2,3]. Osteoclasts, derived from hematopoietic stem cells (HSCs), secrete acids and specialized proteinases to break down and digest complexes of hydrated proteins and minerals at the molecular level in a process known as bone resorption [4,5,6]. Osteoblasts originating from mesenchymal stem cells (MSCs) are involved in osteogenesis and development of the bone matrix, which is formed by tropocollagen, a component of collagenous fiber, and mucopolysaccharide synthesis [7,8,9,10,11]. An imbalance in bone remodeling is involved in several bone-related diseases (for example, osteoporosis, osteopetrosis, osteoarthritis, rheumatoid arthritis, Paget’s disease, osteonecrosis, and scoliosis).

Mammalian-derived exosomes are secreted by almost all cells and have a size of 50–300 nm [12]. Exosomes transport biomolecular cargoes including lipids, proteins, and nucleic acids to target cells for biological activation. Exosomes have been used as markers for specific diseases. Plant-derived nanovesicles are released from various edible plants and have characteristics similar to mammalian-derived exosomes. Many natural biochemicals present in plant-derived nanovesicles can be delivered to target tissues, resulting in enhanced therapeutic effects [13].

Previous reports have shown that plums and dried plums contain abundant amounts of polyphenols and increase bone density, skeletal bone mass acquisition, and indicators of bone formation [14]. Polyphenols from dried plum have also been demonstrated to prevent bone resorption by osteoclasts through suppression of receptor activator NF-kB ligand (RANKL) signaling [15]. The use of dried plums as a dietary supplement can protect bone from deterioration after spinal cord injury [16]. When vitamin K, potassium, and polyphenols from dried plums were combined, trabecular and cortical bone parameters were recovered. [17]. However, the effects of plum-derived exosome-like nanovesicles (PENVs) on osteoblast and osteoclast activation have not been reported to date.

In this study, we have demonstrated that PENVs suppressed osteoclast activation and modulated differentiation and mineralization of osteoblasts for the first time.

## 2. Materials and Methods

### 2.1. Reagents

Murine RANKL (cat No. 315-11) and M-CSF (cat No. 315-02) were purchased from PeproTech. Cell cultures with high quality reagents were used from Gibco (Grand Island, NY, USA). Unless otherwise specified, additional reagents were used from Sigma-Aldrich (St. Louis, MO, USA).

### 2.2. Cell Culture

MC3T3-E1 subclone 4 (#CRL-2593) was obtained from the American Type Culture Collection (ATCC). The cells were grown in growth media [α-MEM containing 10% FBS, penicillin-streptomycin (10,000 U/mL)] and treated osteoblast differentiation media (growth media with 50 μg/mL ascorbic acid and 10 mM sodium phosphate monobasic).

Long bones obtained from 7-week-old female mice were used to produce primary osteoblast cultures by flushing the bone marrow with RPMI medium. Primary osteoblasts from mouse bone marrow were grown in growth media (α-MEM containing Penicillin-Streptomycin (10,000 U/mL) for 24 h. The adherent cells, considered to be stromal cell populations, were trypsinized using 0.25% trypsin-EDTA (Sigma-Aldrich, St. Louis, MO, USA) and seeded. Primary osteoblasts from mice were maintained in the osteoblast differentiation media (growth media with 50 μg/mL ascorbic acid and 10 mM sodium phosphate monobasic).

For preparation of primary osteoclast cultures, bone marrow cells were extracted from the tibia and femur of 7-week-old female mice. The isolated bone marrow cells were cultured in α-MEM containing 10% FBS, 2 mM L-glutamine and penicillin-streptomycin (10,000 U/mL) for 24 h. Cells floating on the medium were then primed with 30 ng/mL M-CSF for 72 h. After 72 h of exposure, the attached cells were considered to be bone-marrow-derived macrophages (BMMs). For differentiation of osteoclasts, the BMMs from the indicated groups were incubated in osteoclast differentiation media (α-MEM medium containing 30 ng/mL M-CSF and 100 ng/mL RANKL) for 72–96 h.

### 2.3. Purification and Isolation of Plum-Derived Exosome-like Nanovesicles

Japanese plums, known as daeseoks, were cultivated on Gyeongsang, Uiseong, and harvested [18]. For isolating PENVs, Japanese plums were deseeded and made into a fresh juice using juicer. Fresh plum juice was used in sequential centrifugation following our lab methods for the isolation of plant exosome-like nanovesicles (Figure 1A) [19]. The PENVs pellet was resuspended in 500 μL PBS and stored at −80 °C. The concentration of PENVs was determined using the Bicinchoninic Acid (BCA) Protein Assay Kit (Thermo Fisher Scientific, Rockford, IL, USA).

### 2.4. Nanoparticle Tracking Analysis (NTA)

The size distribution and particles concentration of the average PENVs were determined by NTA using a Nanosight NS300 device (Malvern Panalysis Ltd., Malvern, UK) [19].

### 2.5. Transmission Electron Microscopy (TEM) and Cryogenic Electron Microscopy (Cryo-EM)

PENVs were applied to low-discharged TEM grid covered with a continuous carbon film for 1 min. Grids were negatively stained with 0.75% (*w*/*v*) uranyl formate, as previously reported [20]. TEM images were recorded with a magnification of 22,000×. Cryo-EM (Titan Krios G4 Cryo-TEM; Thermo Fisher Scientific) was used to observe the PENVs, and the images were captured at a magnification of 22,000× [19].

### 2.6. Labeling of PENVs and Uptake of Labeled PENVs in Cell Cultures

PENVs (1 mg in 1 mL of PBS) were labeled with DiD red fluorescent dye (1,1′-dioctadecyl-3,3,3′,3′-tetramethylindotricarbocyanine iodide; Invitrogen) at 1:200 dilution following to the manufacturer’s instructions. For uptake of PENVs into MC3T3-E1 cells, DiD dye-labeled PENVs were acquired using the CELENA^®^ S Digital Imaging System (Anyang-si, Republic of Korea) for 6 or 24 h [19].

### 2.7. MTT Assay

MC3T3-E1 cells (density of 1 × 10^4^ cells/well in 96-well plates) were treated in a growth medium in the presence or absence of PENVs for the indicated number of days. After incubation of cells, MTT reagent was added to all wells in 96-well plates. The medium in the wells was aspirated, and the wells were refilled with DMSO to dissolve formazan crystals. The purple-colored formazan product was measured at 570 nm using ELISA plate reader (TECAN, Männedorf, Switzerland).

### 2.8. Determination of Alkaline Phosphatase (ALP) Activity

MC3T3-E1 cells (density of 1 × 10^5^ cells/well in 12-well plates) were treated in a differentiation medium in the presence or absence of PENVs for the indicated number of days. The activity of alkaline phosphatase (ALP) was used in our lab method to determine cellular or medium ALP activity in the presence or absence of PENVs [19]. ALP activity in the cells is presented as nmol PNP/mg protein/min and nmol PNP/mL/min as units of enzyme activity.

### 2.9. Alizarin Red S and von Kossa Staining

MC3T3-E1 cells and primary osteoblasts (density of 1 × 10^5^ cells/well in 12-well plates) were treated in a differentiation medium in the presence or absence of PENVs for the indicated number of days. On day 7, MC3T3-E1 cells and primary osteoblasts were washed in PBS and fixed in ethanol. The MC3T3-E1 cells and primary osteoblasts were stained with 40 mM Alizarin Red S solution (pH 4.4) for 30 min at room temperature and washed with PBS. Images of the mineralized matrices were acquired using a microscope (Leica, Nussloch, Germany).

To quantify matrix mineralization, 100 mmol/L cetylpyridinium chloride was dissolved over 1 h at room temperature. The absorbance of ca-deposit with stained red colored nodule was measured at absorbance intensity (570/600 nm) using ELISA plate reader (TECAN, Männedorf, Switzerland) [19].

For von Kossa staining, MC3T3-E1 cells and primary osteoblasts (density of 1 × 10^5^ cells/well in 12-well plates) were treated in a differentiation medium in the presence or absence of PENVs for the indicated number of days. The MC3T3-E1 cells and primary osteoblasts were stained with 3% silver nitrate at room temperature under ultraviolet light for 1 h. Images of the mineralized matrices were captured using a microscope (Leica, Nussloch, Germany) [19].

### 2.10. Tartrate-Resistant Acid Phosphatase (TRAP) Staining and Activity Analysis

The osteoclasts differentiated from mouse bone marrow cells were stained for TRAP, an enzyme found abundantly in mature osteoclasts. TRAP staining (TRAP & ALP double-stain kit, cat no. MK300, Takara, Kusatsu, Japan) was performed according to the manufacturer’s instructions [19]. After washing with PBS, the cells were stained using a TRAP staining kit. TRAP-positive cells with at least three nuclei were regarded as osteoclasts and were detected using a light microscope (Leica Microsystems, Wetzlar, Germany).

For the TRAP activity assay, the osteoclasts were permeabilized with 0.5% Triton X-100 for 10 min and incubated with pNPP substrate-containing solution. The reaction mixtures were transferred to a new 96-well plate containing an equal volume of 1 N NaOH, and the absorbance of TRAP activity was measured at 405 nm using ELISA plate reader (TECAN, Männedorf, Switzerland).

### 2.11. Real-Time Quantitative Polymerase Chain Reaction (RT qPCR)

The mRNA levels of Runx2, osteopontin (OPN), ALP, and collagen type 1 (COL1) were determined using RT qPCR analysis. Total RNA was extracted using the Total RNA Extraction Kit (Thermo Fisher Scientific, Lenexa, KS, USA) and RNA concentration was measured using Nanodrop^®^ ND-1000 (Thermo Fisher Scientific). cDNA was amplified using a cDNA reverse transcription kit (Applied Biosystems, Foster City, CA, USA). To determine the relative mRNA expression, glyceraldehyde 3-phosphate dehydrogenase (GAPDH) with SYBR Green PCR Master Mix (Applied Biosystems) was used as indicated in previous reports [19]. The primer sequences used are listed in Table 1.

### 2.12. Western Blot Analysis

The cellular proteins were extracted using 1 × RIPA buffer from cell pellets and the concentration of cellular proteins was measured using a BCA protein assay kit. Equal amounts of proteins were separated by sodium dodecyl sulfate-polyacrylamide gel electrophoresis (SDS-PAGE) and transferred onto nitrocellulose membranes. After blocking with 5% skim milk in TBS-T (Tris-buffered saline containing 0.1% Tween-20), the membranes were incubated with primary antibodies overnight at 4 °C. The primary antibodies against ALP, OPN, Runx2, Osterix, BMP-2, Smad-1, p38, ERK, JNK, β-actin (1:5000 dilution), phospho-Smad1, phospho-p38, phospho-ERK, phospho-JNK (1:3000 dilution), PPAR gamma, c-Fos, and NFATc1 were used from Santa Cruz Biotechnology (San Diego, CA, USA). Next, the nitrocellulose membranes were primed with secondary antibodies (Anti-Rabbit or Anti-Mouse with horseradish peroxidase). Immunoreactive protein bands were detected using a chemiluminescent reagent (Thermo Fisher Scientific). Relative protein images were captured using Fusion SOLO X (Vilber, Marne-la-Vallée, France).

### 2.13. Statistical Analysis

Data are expressed as means ± SD. Statistical comparisons were performed using one-way analysis or t-test of variance, followed by Tukey’s post hoc analysis. Statistical analyses were performed using the SPSS software (version 21.0; SPSS, Inc., Chicago, IL, USA) [19]. *p* value < 0.5 was considered statistically significant.

## 3. Results

### 3.1. Isolation and Characterization of PENVs

First, we isolated PENVs using an optimized mammalian exosome purification method [19] (Figure 1A). Vesicle structure and size of the PENVs particles were observed using TEM and cryo-electron microscopy (Figure 1B). The size of the PENVs was 211 nm, as measured by Nanoparticle Tracking Analysis (NTA) (Figure 1C). To confirm whether the PENVs were internalized into the cells, we used the DiD labeling technique (Figure 1D). Following 6 and 24 h of treating PENVs with DiD, red fluorescence of the DiD signal was visible in MC3T3-E1 cells but not in control cells (Figure 1E), indicating that PENVs were integrated well into the osteoblastic MC3T3-E1 cells.

**Figure 1 nutrients-15-02107-f001:**
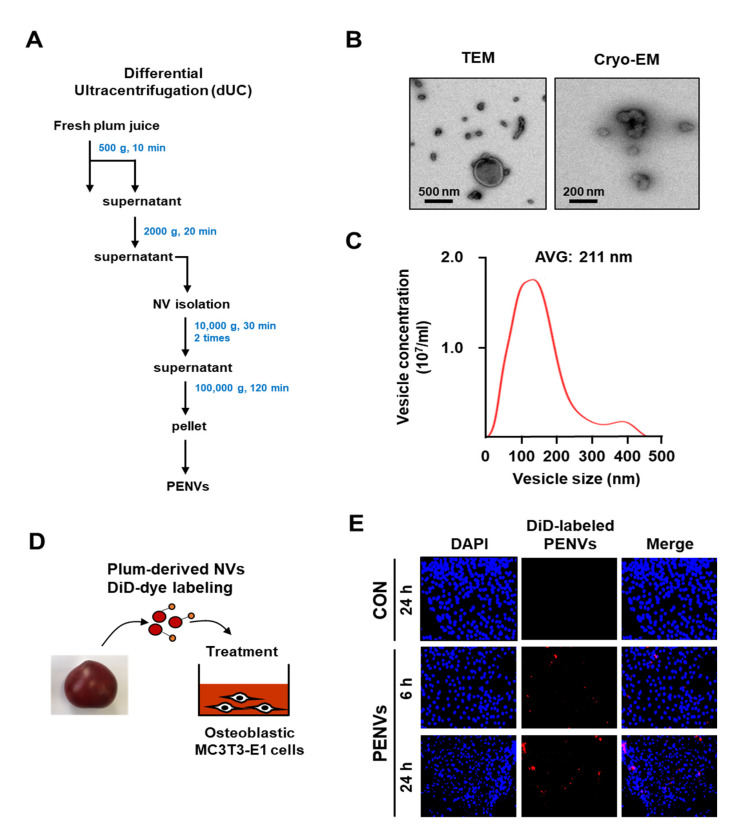
Characterization of plum-derived exosome-like nanovesicles. (**A**) Isolation of PENVs by ultracentrifugation. (**B**) Transmission electron microscopy (TEM) and Cryo-electron microscopy (Cryo-EM) were used to detect the morphology and size of PENVs. (**C**) Particle sizes were measured by Nanoparticle Tracking Analysis. (**D**) Experimental set-up of uptake by PENVs of osteoblasts. (**E**) Fluorescence images of MC3T3-E1 cells treated with DiD-labeled PENVs. Nuclei were stained with DAPI.

### 3.2. PENVs Activated the Osteoblast Differentiation and Mineralization

To determine whether PENVs can modulate osteoblast activation, various doses of osteoblastic MC3T3-E1 cells were treated with PENVs (1, 5, 10, and 50 μg/mL). MTT assay revealed that PENVs did not display toxicity against osteoblastic MC3T3-E1 cell proliferation in a dose- or time-dependent manner (Figure 2A). Treatment with various doses of PENVs for 7 or 14 days increased ALP activity (Figure 2B).

Mineralization of osteoblasts was determined by Alizarin Red S and von Kossa staining, which showed that the treatment with PENVs for 7 days promoted the deposition of Ca-P in the MC3T3-E1 cells (Figure 2C–E). Thus, our data demonstrated that differentiation and mineralization of osteoblasts were modulated by PENVs.

### 3.3. PENVs Enhanced Osteoblast Differentiation Marker Gene and Protein Expressions

ALP and OPN play critical roles in bone metabolism by activating osteoblast differentiation [21,22]. Osterix and Runx2 are essential transcription factors that modulate osteoblast differentiation [23,24]. To investigate whether PENVs modulate MC3T3-E1 cell differentiation, osteoblast differentiation transcription factors and marker genes were analyzed using qRT-PCR. Osteoblast differentiation marker genes, including ALP, OPN, COL1, and osteocalcin (OCN), and transcription factors, such as Osterix and Runx2, were elevated in cells treated with PENV for 7 d (Figure 3A).

To confirm whether PENVs can activate osteoblast differentiation marker proteins, including ALP, OPN, Runx2 and Osterix, we examined the expression levels of these proteins in MC3T3-E1 cells using Western blot analysis. The results showed that the expression of osteoblast differentiation marker proteins (ALP and OPN) and transcription factor proteins (Runx2 and Osterix) in PENV-treated cells was higher than in PENV-untreated cells (Figure 3B,C). Overall, these results indicate that PENVs can modulate the expression of osteoblast differentiation markers and transcription factors related to genes and proteins.

### 3.4. PENVs Stimulated Osteoblast Differentiation through BMP-2/MAPK/Smad-1 Dependent Runx2 Pathway

Bone morphogenetic protein-2 (BMP-2) is a key signaling target molecule in osteoblast differentiation and formation [25,26]. BMP-2 is an upstream regulator of the Runx2, an osteoblast differentiation transcription factor. Additionally, BMP-2 modulates the activation of the major mitogen-activated protein kinase (MAPK) molecules (p38, JNK, and ERK) and Smad-1 activation. To determine whether PENVs promote BMP-2 signaling, we performed RT qPCR to determine the mRNA levels of *BMP-2* and *Smad-1*, followed by Western blot analysis to confirm the expression of phospho-Smad1, -p38, -ERK and -JNK. The gene expression levels of *BMP-2* and *Smad-1* were enhanced in PENV-treated osteoblastic MC3T3-E1 cells (Figure 4A). In addition, PENVs markedly increased the phosphorylation of JNK, p38, and Smad1 in MC3T3-E1 cells (Figure 4B). However, the PENVs did not modulate the phosphorylation of the ERK protein. These results suggest that PENVs enhance osteoblast differentiation genes and proteins through the BMP-2/MAPK/Smad-1 dependent Runx2 pathway (Figure 4C).

### 3.5. PENVs Enhanced Osteoblast Differentiation and Mineralization of Mouse Primary Osteoblasts

To confirm primary osteoblast differentiation and mineralization induced by PENVs, we performed ALP, Alizarin Red S, and von Kossa staining. ALP staining showed that PENV-treated mouse primary cells exhibited strong ALP-positive staining in a dose-dependent manner (Figure 5A). Alizarin red S and von Kossa staining also showed high expression of Ca-P deposits in mouse primary cells treated with various doses of PENVs-treated mouse primary cells for 7 days (Figure 5B,C). The intensity of Alizarin Red S staining in the nodules with mineral deposits increased with various doses of mouse primary cells treated with PENV for 7 days (Figure 5D). Taken together, these data indicate that PENVs promote osteoblast differentiation and mineralization of primary osteoblasts, similar to the osteoblastic MC3T3-E1 cell line.

### 3.6. PENVs Inhibited Osteoclast Differentiation of Mouse Primary Osteoclasts

During bone remodeling, osteoclasts are inextricably linked to osteoblasts [27]. Thus, we tested whether PENVs were able to regulate osteoclasts rather than osteoblasts. TRAP+ cells decreased in the PENV-treated group (Figure 6A). Consistent with these results, PENVs significantly decreased TRAP activity and osteoclastogenic differentiation of marker of proteins such as NFATc1, c-Fos, and PPAR-gamma only at the concentration of 50 μg/mL (Figure 6B,C). Our data showed that PENVs efficiently promote bone formation by reducing osteoclast differentiation.

## 4. Discussion

In the current study, we first isolated and characterized exosomes, such as nanovesicles, extracted from fresh plum juice. We found that PENVs exhibited biological functions in the regulation of osteoblasts and osteoclasts activation (Figure 7). Treatment with PENVs enhanced osteoblast activation in osteoblastic MC3T3-E1 cells and primary osteoblasts in mice. In addition, PENVs increased the mRNAs and protein expression of osteoblast differentiation markers, such as ALP and OPN, and transcription factors, such as Osterix and Runx2. In particular, activation of the BMP-2/MAPK/Samd-1 dependent Runx2 pathway was induced by treatment with PENVs, suggesting that PENVs promote osteoblast differentiation through BMP-2/MAPK/Samd-1 molecules. In addition, PENVs treatment inhibited osteoclastogenesis in primary mouse osteoclasts. Therefore, these results imply that PENVs could be used for the optimal treatment of osteoporosis.

Exosomes are small vesicles composed of vesicles containing various proteins, nucleic acids, bioactive lipids, and secondary metabolites, which act as extracellular messengers between various cells [28,29,30]. Plant-derived exosomes, such as nanovesicles, also contain similar molecules. Interestingly, these nanovesicles have an advantage of biocompatibility and biodegradability, making them suitable as vehicles for therapeutic delivery, similar to mammalian exosomes [31,32]. Several studies have reported that plant-derived nanovesicles target some types of tissues and are involved in the prevention of inflammation or oxidative stress in various diseases. Recently, our group reported that yam-derived nanovesicles are important in prevention of osteoporosis in vitro and in vivo [19]. We also reported that apple-derived nanovesicles regulated osteoblastic MC3T3-E1 cell line activation [33].

Polyphenols in plums and prunes have been reported to significantly restore bone mass by upregulating osteoblast function and activity [14]. Another study showed that dried plums have preventive effects against bone loss caused by ovariectomy in C57BL/6J mice [34] and bone-related diseases in humans [35]. Focusing on the protective efficacy of plums against bone deterioration, we hypothesized that PENVs promote osteoblast differentiation and suppress osteoclast activation.

Osteoblasts, which are derived from mesenchymal stem cells, form bones, and together with osteoclasts and osteocytes are responsible for bone resorption, constitute the bone tissue that maintains its strength and elasticity [36]. Osteoblast activation leads to a reduced incidence of bone-mass-related diseases [37]. To promote and strengthen bone formation, an increase in the activation and differentiation of osteoblasts and/or a decrease in death of osteoblasts is important. Our data showed that both cellular ALP activity and ALP-stained positive areas increased in the osteoblastic cell line MC3T3-E1 and primary osteoblasts obtained from mice. The expression of osteoblast differentiation marker genes, such as ALP and OPN, and osteoblast transcription factors, such as Runx2 and Osterix, was significantly elevated in PENV-treated osteoblastic MC3T3-E1 cells. PENVs effectively increased the expression levels of the osteogenic genes as well as those of proteins. In addition, mineralized bone nodule formation was increased after treatment of osteoblastic MC3T3-E1 cells and primary osteoblasts obtained from mice.

BMP-2 plays a major role in osteoblast differentiation and commitment. In addition, among the several signaling pathways regulating Runx2, such as those of transforming growth factor (TGF2) and fibroblast growth factor (FGF), the BMP-2 signaling pathway is the most well-known [25,26]. In addition, BMP-2 activates major families of MAPKs, such as ERK, p38, JNK, and Smad, and subsequently induces the expression of Runx2, an important transcription factor involved in the differentiation of osteoblastic cells. Our data showed that treatment with PENVs increased the expression of BMP-2 gene and protein in MC3T3-E1 cells. In addition, treatment with PENVs markedly elevated the expression of phospho-Smad1, phospho-p38, and phospho-JNK but did not affect the expression of phospho-ERK (Figure 4). These results demonstrate that modulation of osteoblast differentiation/proliferation by treatment with PENVs is closely related to the Runx2-dependent BMP-2/MAPK/Smad-1 signaling pathway.

Osteoclasts play a major role in maintaining bone remodeling [38]. However, an increase in osteoclasts or their hyperactivation may cause an imbalance in bone homeostasis, thereby contributing to the development of bone-related diseases, such as osteoporosis, bone fracture, and bone loss. In the present study, PENVs suppressed RANKL-dependent osteoclast formation and differentiation without having any cytotoxic effects. Treatment with PENVs prior to RANKL treatment significantly inhibited RANKL-induced formation of TRAP-positive cells (Figure 6A). In addition, TRAP activity and osteoclast activation of PPAR-gamma, NFATc1 and c-Fos proteins were inhibited by treatment only with 50 µM of PENVs (Figure 6). Thus, PENV treatment significantly inhibited RANKL-induced osteoclast differentiation.

In summary, this study demonstrates that PENVs enhance osteoblast differentiation and mineralization by modulating the BMP-2/MAPK/Smad-1 dependent Runx2 pathway. We also confirmed that PENVs promote differentiation and mineralization of primary osteoblasts. Regarding osteoclast differentiation markers, PENVs inhibited TRAP+ cells and TRAP activity in mouse primary osteoclasts. To the best of our knowledge, these results are the first to report that PENVs can be used as osteoporosis treatment to promote osteoblast activation and decrease osteoclast differentiation.

## Figures and Tables

**Figure 2 nutrients-15-02107-f002:**
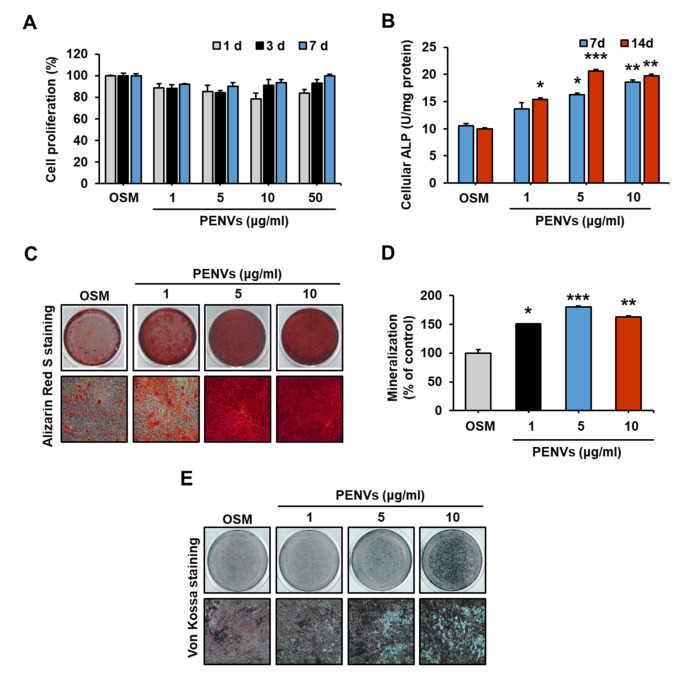
PENVs modulate osteoblast activation and differentiation. (**A**) Proliferation of MC3T3-E1 cells is shown using MTT assay. (**B**) Cellular ALP activity in osteoblastic MC3T3-E1 cells treated with PENVs. (**C**) Representative images of Alizarin Red S staining of MC3T3-E1 cells after treatment with PENVs for 7 days are shown. (**D**) Ca deposit in extracellular matrix was quantified using Alizarin Red S dye. (**E**) Representative images of von Kossa staining after treatment of MC3T3-E1 cells with PENVs for 7 days. * *p* < 0.05, ** *p* < 0.01, *** *p* < 0.001 between PENVs and control group (OSM; osteogenic medium).

**Figure 3 nutrients-15-02107-f003:**
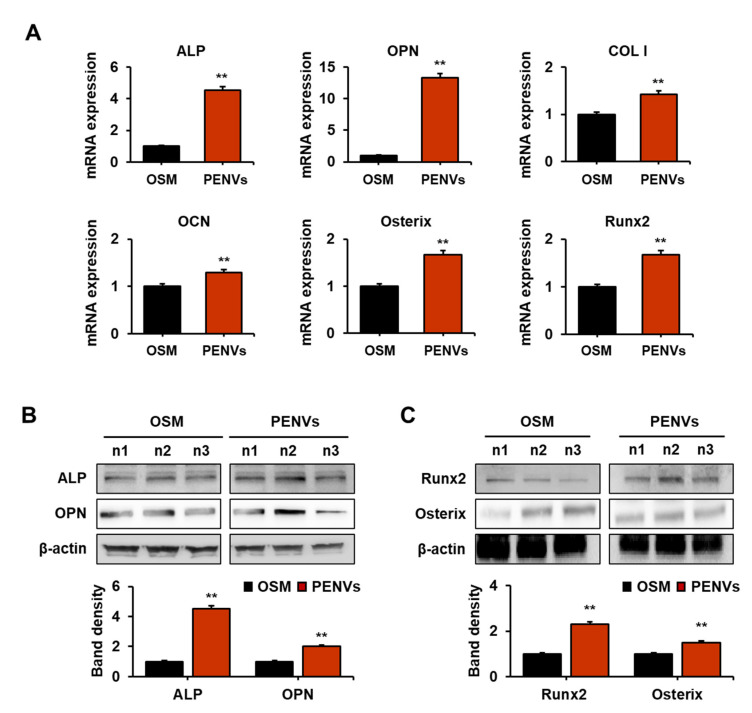
PENVs promote the expression of mRNA and protein of osteoblast differentiation markers. (**A**) mRNA expression of osteoblast markers was determined by real-time qPCR. (**B**) Levels of alkaline phosphate (ALP) and osteopontin (OPN) were determined using Western blot analysis. (**C**) Expressions of Osterix and Runx2 as osteoblast differentiation transcription factors in osteoblastic MC3T3-E1 cells are shown. Data are presented as means ± SD (*n* = 3). ** *p* < 0.01 between PENVs and control group (OSM; osteogenic medium).

**Figure 4 nutrients-15-02107-f004:**
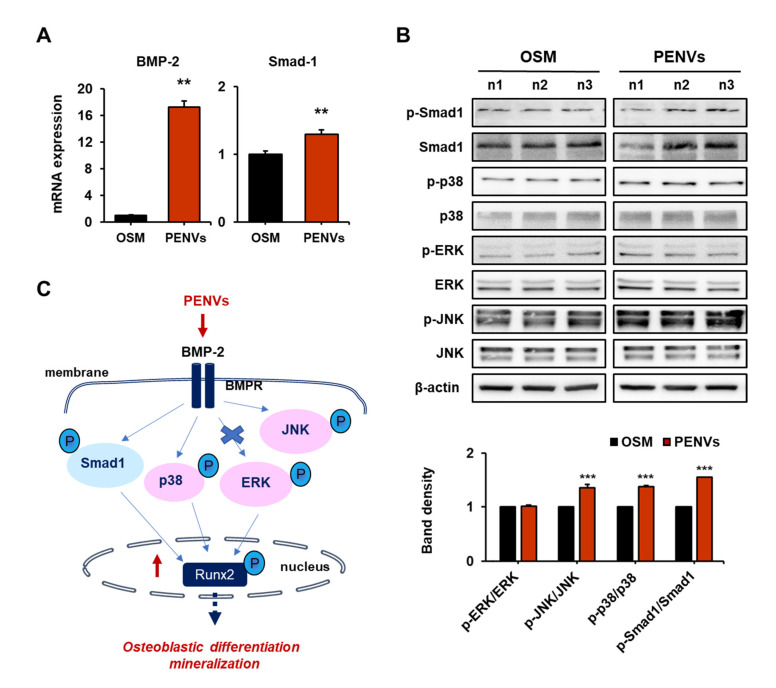
PENVs modulate osteoblast differentiation through BMP-2 signaling. (**A**) mRNA levels of BMP-2 and Smad-1 using Real-time quantitative polymerase chain reaction (RT qPCR). (**B**) Representative Western blot bands of different proteins are shown. (**C**) A schematic illustration depicting the effects of PENVs in promoting osteoblast differentiation. Data presented as means ± SD (*n* = 3). ** *p* < 0.01, *** *p* < 0.001 between PENVs and control group (OSM; osteogenic medium).

**Figure 5 nutrients-15-02107-f005:**
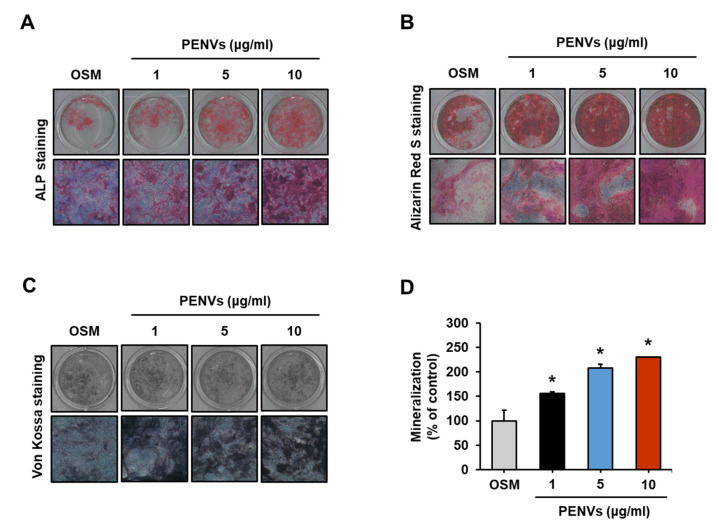
PENVs enhance differentiation of mouse primary osteoblasts. (**A**–**C**) Representative images of ALP staining, Alizarin Red S staining, and von Kossa staining are shown. (**D**) Ca deposition in extracellular matrix was quantified using Alizarin Red S dye. Data presented as the means ± SD (*n* = 3). * *p* < 0.05 between PENVs and control group (OSM; osteogenic medium).

**Figure 6 nutrients-15-02107-f006:**
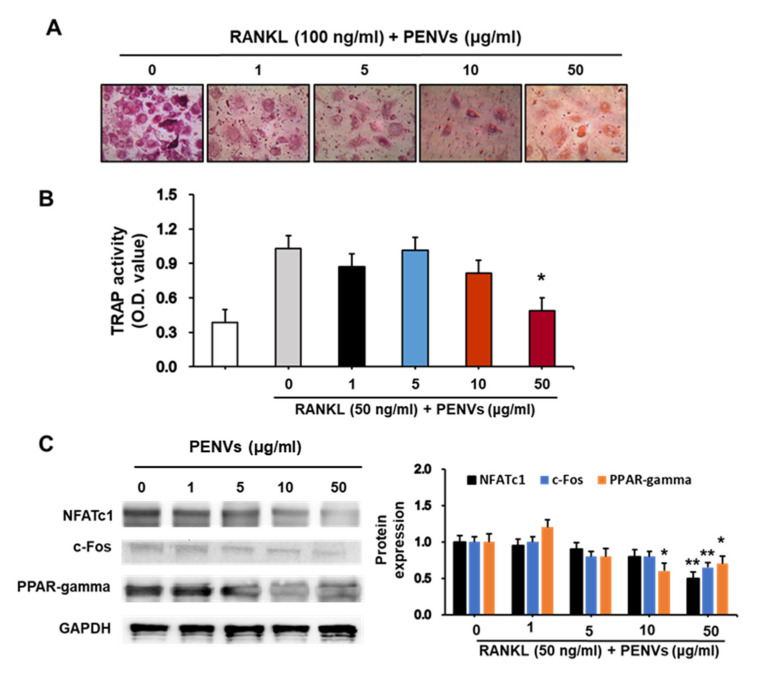
PENVs decrease osteoclast activation in mouse primary osteoclasts. Bone marrow cells from 7-week-old mice were isolated and incubated for 3 days in growth medium containing M-CSF. The cells were considered to be differentiated into macrophages and used in the osteoclast study as mentioned in “Materials and Methods” section. After stimulation with RANKL, (**A**) TRAP staining and (**B**) TRAP activity (absorbance at 405 nm) was examined. Data presented as the means ± SD (*n* = 3). (**C**) Levels of the osteoclastogenesis proteins were estimated *(n* = 3). * *p* < 0.05, ** *p* < 0.01 between PENVs and control group (OSM; osteogenic medium).

**Figure 7 nutrients-15-02107-f007:**
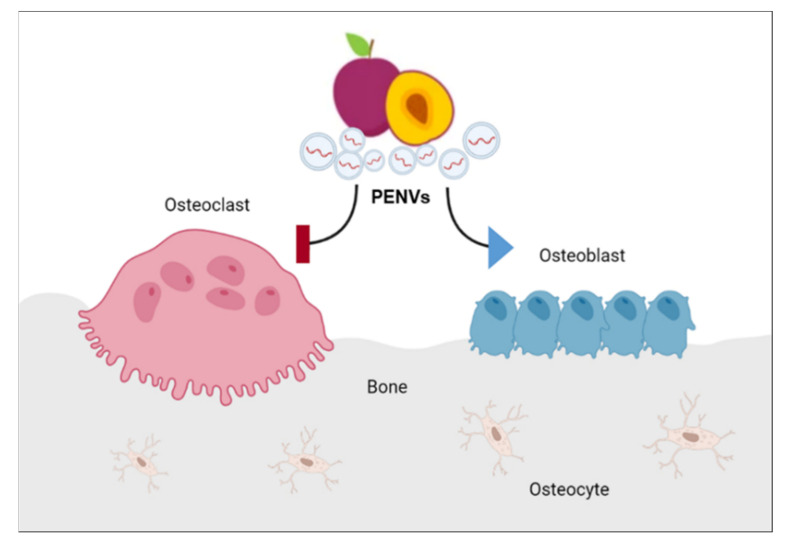
A model showing the mechanism of action of PENVs in promoting bone remodeling by enhanced osteoblast differentiation and suppressed osteoclast activation in osteoblasts and osteoclasts.

**Table 1 nutrients-15-02107-t001:** Primer sequences used for real-time qPCR analysis.

Genes	Sequence (5′ → 3′)
ALP	F: CAAGGATGCTGGGAAGTCCG R: CGGATAACGAGATGCCACCA
OPN	F: CTGGCAGCTCAGAGGAGAAG R: CAGCATTCTGTGGCGCAAG
COL 1	F: ACGTCCTGGTGAAGTTGGTC R: CAGGGAAGCCTCTTTCTCCT
OCN	F: GCAATAAGGTAGTGAACAGACTC R: GTTTGTAGGCGGTCTTCAAGC
Osterix	F: GTCAAGAGTCTTAGCCAAACTC R: AAATGATGTGAGGCCAGATGG
BMP-2	F: CGCACGCGATGCAACACCAC R: ACTGCATGTCCCCGGGCTCA
Smad-1	F: AAGGTGGGGAAAGTGAAAC R: CTGCTTGGAACCAAATGGGAA

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
