# Peer review of "Plum-Derived Exosome-like Nanovesicles Induce Differentiation of Osteoblasts and Reduction of Osteoclast Activation"

_nutrients, 2023, doi:10.3390/nu15092107_

Round 1

Reviewer 1 Report

The manuscript is very interesting, and it was a pleasure reading it. However, some aspects should be improved.

1- Figure 2: The statistical analysis should be more understandable;

2- 3.4. PENVs stimulated osteoblast differentiation through BMP-2/MAPK/Samd-1 dependent Runx2 pathway. You have to modify Samd-1 in Smad-1;

3- Specify each abbreviations used on your first use (i.e. BMP-2);

4- Figure 5D: The statistical analysis should be more understandable;

5- Several references are old. Could you replace them with equally authoritative and more recent sources?

Author Response

Point-by-Point response to each of the reviewers’ comments (nutrients-2297206):

Reviewer 1

Comments and Suggestions for Authors

The manuscript is very interesting, and it was a pleasure reading it. However, some aspects should be improved.

1- Figure 2: The statistical analysis should be more understandable.

Response: We appreciated the reviewer's comment. Therefore, we changed it.

2- 3.4. PENVs stimulated osteoblast differentiation through BMP-2/MAPK/Samd-1 dependent Runx2 pathway. You have to modify Samd-1 in Smad-1.

Response: We appreciated the reviewer's comment. To respond to this reviewer’s comment, we changed the manuscript.

3- Specify each abbreviations used on your first use (i.e. BMP-2).

Response: We appreciated the reviewer's comment. Therefore, we added the full term the first time when we used the abbreviations term.

4- Figure 5D: The statistical analysis should be more understandable.

Response: We appreciated the reviewer's comment. Therefore, we changed it.

5- Several references are old. Could you replace them with equally authoritative and more recent sources?

Response: We understand the reviewer’s excellent comments. Therefore, we replaced it.

Reviewer 2 Report

Comment to Authors

This manuscript entitles “Plum-derived exosome-like nanovesicles induce the differentiation of osteoblasts and reduction of osteoclast activation” is an original research article with 37 references. The main text was divided into four main sections, including the introduction, materials and methods, results, and discussion (including the conclusion in the last paragraph).

Brief summary: The study focuses on the effect of plum-derived exosome-like nanovesicles (PENVs) on the osteoblast and osteoclast activities, as no previous finding has been reported. Plant-derived exosomes like nanovesicles have the advantages of biocompatibility and biodegradability, making them suitable for use as a vehicle for therapeutic delivery, similar to mammalian exosomes. Findings indicated that PENVs significantly affect markers protein for the MCT3T-E1 cell line, marked by a significant increase in ALP, OPN, Runnx2, Osterix, BMP-2, and Smad-1 as well as in primary osteoblast from a mouse that showed a significant increase in osteoblast differentiation and mineralization. Significant decreases in osteoclast differentiation indicated by TRAP activity of primary mouse cells were also observed.

Comment 1: Identifying PENVs activities on osteoblast differentiation and mineralization was using multiple doses (1, 5, 10, and 50 mg/ml), some of which showed significantly increased. Any reason why the authors select only 10 ug/ml for other markers analysis? Why did you choose that concentration?

Comment 2: The authors can conclude which concentration showed the best effects for osteoblast and osteoclast activities. Because only 50 mg/ml was shown significantly decrease the TRAP markers. Any reason for varying concentrations for both cells? Why did the authors not measure osteoclast-related proteins like SSPR, RANKL, PPARG, etc. expression as it might strengthen the findings to support the effects of PENVs on osteoclasts?

Comment 3: A little bit confused about the superscripts letter for a significant difference. Make it clearer which group shows a significantly different compared to which group.

Comments 4: Authors can discuss some of the limitations found in the study and future studies/directions from your findings. Can put the conclusion in another section.  

Author Response

Point-by-Point response to each of the reviewers’ comments (nutrients-2297206):

Reviewer 2

Comment to Authors

This manuscript entitles “Plum-derived exosome-like nanovesicles induce the differentiation of osteoblasts and reduction of osteoclast activation” is an original research article with 37 references. The main text was divided into four main sections, including the introduction, materials and methods, results, and discussion (including the conclusion in the last paragraph).

Brief summary: The study focuses on the effect of plum-derived exosome-like nanovesicles (PENVs) on the osteoblast and osteoclast activities, as no previous finding has been reported. Plant-derived exosomes like nanovesicles have the advantages of biocompatibility and biodegradability, making them suitable for use as a vehicle for therapeutic delivery, similar to mammalian exosomes. Findings indicated that PENVs significantly affect markers protein for the MCT3T-E1 cell line, marked by a significant increase in ALP, OPN, Runnx2, Osterix, BMP-2, and Smad-1 as well as in primary osteoblast from a mouse that showed a significant increase in osteoblast differentiation and mineralization. Significant decreases in osteoclast differentiation indicated by TRAP activity of primary mouse cells were also observed.

Comment 1: Identifying PENVs activities on osteoblast differentiation and mineralization was using multiple doses (1, 5, 10, and 50 mg/ml), some of which showed significantly increased. Any reason why the authors select only 10 ug/ml for other markers analysis? Why did you choose that concentration?

  • Because 10mg/ml was more effectfully differentiation then 1, 5mg/ml on osteoblast. And 50mg/ml makes saturation of the osteoblast differentiation.

Response:  We appreciated the reviewer's comment. We tested 1, 5, and 10 ug/ml of PENVs for evaluating both osteoblast differentiation and mineralization in osteoblastic MC3T3-E1 cells. Honestly, we found that 10 ug/ml of PENVs was significantly elevated in osteoblast differentiation and mineralization of osteoblastic MC3T3-E1 cells. Thus, we used 10 ug/ml of PENVs for confirmation of BMP-2 mechanism.

Comment 2: The authors can conclude which concentration showed the best effects for osteoblast and osteoclast activities. Because only 50 mg/ml was shown significantly decrease the TRAP markers. Any reason for varying concentrations for both cells? Why did the authors not measure osteoclast-related proteins like SSPR, RANKL, PPARG, etc. expression as it might strengthen the findings to support the effects of PENVs on osteoclasts?

Response: We appreciated the reviewer's comment. We added the image on Figure 6C. We found that PENVs significantly decreased TRAP activity and ostoeclastgenesis such as PPAR-gamma, NFATc1, and c-Fos proteins at only 50 μg/ml concentration.

Comment 3: A little bit confused about the superscripts letter for a significant difference. Make it clearer which group shows a significantly different compared to which group.

Response: We appreciated the reviewer's comment. Therefore, we changed it.

Comments 4: Authors can discuss some of the limitations found in the study and future studies/directions from your findings. Can put the conclusion in another section.

  • In this study we confirm the effect of PENVs(plum-derived exosome-like nanovesicles) on osteoblast. However, we have to know about plant-derived nanovesicle’s clinical efficiency, stability, safety and others.

Response: We understand the reviewer’s excellent comments. Therefore, we modified in this manuscript.